# Correlation Between Catheter Ablation Timing and the Duration of Atrial Fibrillation History on Arrhythmia Recurrence in Patients with Paroxysmal Atrial Fibrillation: A Systematic Review and Meta-Analysis

**DOI:** 10.3390/jcm14196995

**Published:** 2025-10-02

**Authors:** Obaida Makdah, Feras Al Krayem, Cosmin Gabriel Ursu, Mohamad Hussam Sahloul, Oana Gheorghe-Fronea, Radu Vătãsescu, Dan L. Musat, Ștefan Bogdan

**Affiliations:** 1Faculty of Medicine, Carol Davila University of Medicine and Pharmacy, 050474 Bucharest, Romania; firaskrayem2@stud.umfcd.ro (F.A.K.); cosmin.d.v.ursu@stud.umfcd.ro (C.G.U.); hussam-sahloul.mohamad@stud.umfcd.ro (M.H.S.); oana.fronea@umfcd.ro (O.G.-F.); bogdan.stefan@umfcd.ro (Ș.B.); 2Emergency Clinical Hospital, 014461 Bucharest, Romania; 3Valley Health System, Paramus, NJ 07652, USA; musada@valleyhealth.com; 4Elias Clinical Emergency Hospital, 011461 Bucharest, Romania

**Keywords:** atrial fibrillation, catheter ablation, early CA, delayed CA

## Abstract

**Background:** Atrial fibrillation (AF) is the most common sustained arrhythmia. AF catheter ablation (CA) is superior to antiarrhythmic drugs (AAD) therapy in maintaining sinus rhythm. However, not much is known regarding the optimal timing of the ablation. **Methods**: A comprehensive literature search was conducted using PubMed, Embase, and Scopus, focusing on studies published from 2013 until 2022 and including both observational studies and randomized controlled trials (RCTs) with patients undergoing ablation for symptomatic paroxysmal or persistent AF using radiofrequency, cryoablation, or both approaches, studies that reported diagnosis-to-ablation time (DAT), a follow-up period, AF recurrence, or AF burden. Studies that included a surgical ablation, a hybrid ablation approach, or an ablation for arrhythmias other than AF were excluded. Left atrial diameter and ejection fraction (EF) were assessed. **Results**: Ten studies were selected out of 1387 identified records. After a follow-up period of one year, the early ablation subgroup had a lower mean AF recurrence rate (29.8%) compared to that of the delayed ablation subgroup (39.5%). The median AF recurrence rate was in the radiofrequency ablation group (44.5%), in the cryoablation group (27.3%). In studies that included paroxysmal AF patients exclusively, the AF recurrence rate was directly proportional to the DAT. **Conclusions**: Our results suggest that DAT correlates with a recurrence rate at one year following AF CA, and that the shorter the DAT the better the outcome, particularly in paroxysmal AF population.

## 1. Introduction

Atrial fibrillation (AF), the most common sustained arrhythmia [1], is a complex condition. It is supraventricular in origin, characterized by an irregularly irregular ventricular response. The mechanism of AF, though not fully understood, is multifactorial, requiring triggers and an underlying substrate that perpetuates the arrhythmia. In its course, AF is an electrical disease characterized by rapidly firing ectopic foci from the pulmonary veins [2]. At this stage, AF is paroxysmal, self-terminating, and, in most cases, within 48 h, but some episodes may last up to 7 days. AF eventually leads to structural remodeling characterized by atrial scarring and fibrosis [3]. Once these modifications have occurred, AF becomes a structural disease, and the episodes of AF become more persistent [4]. Persistent AF (PsAF) lasts longer than seven days. AF is further classified into long-standing persistent AF and continuous AF lasting more than one year. Permanent AF is when AF is finally accepted by both the patient and the physician, and rhythm-control strategies are no longer pursued. Restoration and maintenance of sinus rhythm involve using catheter ablation (CA) or antiarrhythmic drugs (AADs). The main principle of AF CA involves pulmonary vein isolation (PVI). AF CA is superior to AAD therapy in maintaining sinus rhythm. However, not much is known regarding the optimal timing of the ablation. Evidence in recent years suggests that the time between first diagnosis of AF and ablation, or diagnosis-to-ablation time (DAT), impacts AF recurrence rates post-ablation. Observational studies suggest that increasing the DAT and delaying the ablation are associated with worse outcomes post-ablation.

### 1.1. When to Consider Catheter Ablation Indications

The selection of patients for AF ablation is a critical aspect of AF care, involving a shared decision-making process between the patient and physician. Success rates of ablation are influenced not only by procedural technique but also by patient characteristics. The current guidelines advise that CA should be considered only for symptomatic individuals who have either paroxysmal or PsAF and have not responded well to or cannot tolerate AADs [5]. The reason for this is that the landmark CABANA (Catheter Ablation vs. Antiarrhythmic Drug Therapy for Atrial Fibrillation) RCT published in 2019 showed no significant reduction in all-cause mortality or stroke with AF CA compared to AAD [6]. No RCT has shown that AF CA significantly reduces clinical outcomes like stroke and all-cause mortality. This recommendation is particularly relevant for those who continue to experience bothersome symptoms despite medication. Furthermore, CA might even be contemplated as a first-line treatment for carefully selected symptomatic patients in certain cases. These symptoms may include palpitations, shortness of breath, fatigue, chest discomfort, dizziness, reduced exercise capacity, and anxiety. Despite attempts at managing symptoms with medications, these individuals continue to suffer from the effects of AF, impacting their daily life and well-being.

### 1.2. Atrial Fibrillation Subgroups

It is important to note that there is a subgroup of AF patients for whom AF CA has been shown to reduce all-cause mortality. The RCT CASTLE AF [7], as well as a subgroup analysis of the CABANA RCT [8], has shown that AF CA reduces all-cause mortality in patients with heart failure. This is likely because patients with heart failure benefit from the 20% increase in cardiac output from the atrial kick when they are in sinus rhythm. The recently published CASTLE-HTX has shown that ablation in patients with end-stage heart failure with an ejection fraction <35% was associated with a lower likelihood of a composite of death from any cause than medical therapy [9].

### 1.3. Catheter Ablation Is Superior to Antiarrhythmic Drugs

CA has been documented to be superior to treatment by AADs in terms of sinus rhythm maintenance; numerous randomized controlled trials in drug-free patients have shown that ablation outperforms medical therapy for symptom management [10,11,12]. These studies, however, do not rule out initial medication therapy followed by ablation if and when AF symptoms worsen. Andrade et al. in a recent RCT found that patients who were followed for at least three years, all of whom had loop recorders implanted, had a 7.4% higher risk of a first episode of persistent AF in the medical therapy group than in the ablation group (1.9%) [13,14].

## 2. Materials and Methods

### 2.1. Search Strategy

A comprehensive literature search was conducted using PubMed, Embase and Scopus, with the keywords: atrial fibrillation, sinus rhythm, catheter ablation, arrhythmia recurrences, antiarrhythmic drugs, paroxysmal atrial fibrillation, persistent atrial fibrillation, atrial fibrillation burden, diagnosis-to-ablation time, arrhythmia-free survival. The search strategy was designed to identify relevant studies comprehensively, and all retrieved articles were screened according to predefined inclusion and exclusion criteria. This approach ensured broad coverage of the literature and reduced the risk of selective reporting. The databases were last searched in July 2023.

Articles were published in the following papers: The New England Journal of Medicine, The American Journal of Cardiology, EP Europace, Journal of the European Heart Rhythm Association (EHRA), European Heart Journal, The International Journal of Cardiology, and JAMA Cardiology.

All four investigators used Microsoft Excel to summarize the data collected from each article. At first, they did so independently, but then all reviewers took part in four-week meetings and decided if those studies respected the inclusion and exclusion criteria. Microsoft Excel was used to exclude duplicate articles and create the tables presented. This study protocol was not registered in PROSPERO primarily due to timing and the retrospective nature of the project.

### 2.2. Inclusion and Exclusion Criteria

Studies in our systematic review were included based on the following criteria: (1) observational studies or randomized control trials that included patients with symptomatic paroxysmal or PsAF who underwent ablation using either radiofrequency or cryoballoon approach. (2) Studies that reported the timing between AF diagnosis and the ablation procedure. (3) Studies that included a follow-up period. (4) Studies that evaluated AF burden or AF recurrence. All publications were limited to those involving adults. Studies that included a surgical ablation or a hybrid ablation approach were excluded. Studies that included ablation for arrhythmias other than AF were excluded. Ten articles were included in this systematic review. See Figure 1 for the PRISMA study selection flow diagram.

### 2.3. Data Extraction

In this review, we have focused on studies published from 2013 to 2023, encompassing observational and randomized controlled trials (RCTs). These studies primarily investigate the use of three different ablation strategies: radiofrequency (RF) ablation, cryoablation, or a combination of RF and cryoablation. Patient attributes included the age and sex of patients, the total number of patients involved, the type of AF, and the diagnosis-to-ablation time. The diagnosis-to-ablation time (DAT) in these studies ranges from 5 months to 5 years. Extracted data regarding study characteristics involved title, authors, year of publication, study design, follow-up time, ablation strategies, methodology for follow-up, and key parameters like AF recurrence, left atrium diameter, ejection fraction (EF), hypertension, diabetes, heart failure/congestive heart failure, hyperlipidemia/dyslipidemia, CHA_2_DS_2_-VASc score and Obesity/BMI.

Regarding the methodology for follow-up, most of the included studies employ loop recorders and Holter monitoring. The average or median follow-up duration across these studies is approximately one year. We examine the occurrence of AF recurrence following ablation, changes in the size of the left atrium, evaluated as a potential outcome measure, and alterations in EF as an indicator of cardiac function. Also, each study defines its primary outcomes, and we consider these primary endpoints integral to our analysis.

### 2.4. Risk of Bias

Three reviewers independently assessed the risk of bias, with one reviewer adjudicating using the ROB-2 tool [15]. We assessed bias in five domains (randomization process, missing outcome data, deviations from intended interventions, measurement of the outcome, and selection of the reported results) plus overall risk of bias to classify each trial. Discrepancies in evaluating the risk of bias were resolved through discussions within the study team, leading to a final consensus before including the information.

While there are some concerns, given the observational nature of some studies and the potential for undertreating AF with AADs, we determined that the results are trustworthy and can be safely used to support our conclusions. See Figure 2. Risk-of-bias domains: ROB-2 ROB-2: Risk-of-bias tool.

### 2.5. Statistical Analysis

Statistical analyses were performed using R software (version 3.1), employing the meta and metafor packages. Due to the expected heterogeneity among study populations, a random-effects model was used. The between-study variance was estimated using the DerSimonian–Laird method. Proportions were transformed using the arcsine method to stabilize variances, and the pooled proportions were then back-transformed for better interpretability. The ion coefficient was used with the associated scatter plot as a graphical representation. Heterogeneity was assessed qualitatively by comparing study characteristics and quantitatively using the I^2^ statistic and Cochran’s Q. Where feasible, results from the included studies were combined for each outcome to provide an overall estimate of treatment effect. In cases of high heterogeneity, a leave-one-out sensitivity analysis was conducted. If the heterogeneity could not be explained, subgroups were created based on procedural and study characteristics.

## 3. Results

The DAT varied between one month and four years and correlated with AF recurrence and burden. The study’s follow-up was between one and five years, and follow-up methods were symptom review, ECG, Holter monitoring, and event recording, given the degree of clinical heterogeneity seen in participant selection, interventions, and comparators across studies. In order to examine the impact of DAT, meta-regression was also performed as a categorical covariate (<3 vs. ≥3 years) on study effect size in each outcome.

### 3.1. Patient’s Characteristics

There were a total of 10 studies with a total of 4631 patients. The sample sizes in these studies range from 89 to 1241 patients, capturing a broad spectrum of patient populations. The mean age of patients varied between 50.5 and 67.7 years, male (66.2%). See Table 1.

### 3.2. Delayed CA Studies (DAT of Three Years or More)

Delayed CA studies had a DAT of three years or more. In this category, we found five articles by Bunch et al. [16], Kuck et al. [12], Bisbal et al. [17], Hussein et al. [18], and Maurizio et al. [19], varying between observational prospective and retrospective studies alongside clinical trials, with 2872 patients evaluated. Three studies included radiofrequency ablation as an ablation type [12,16,18], while [17] mentioned cryoablation alongside radiofrequency ablation as the operational type. All articles investigated paroxysmal AF, except for [18] PsAF, while [16,17] also had patients with PsAF. The mean DAT was 43.41 months. AF recurrence had an average of 41.6% of the recurrence rates mentioned in the studies. Left atrial diameter was assessed in all studies except for [16], which ranged between 40.8 and 42.7 mm, being around normal 41 mm. Meanwhile, EF was reported in four out of five studies, ranging between 52.3% and 61.2%, respectively [16,17,18,19]. In conclusion, we did not find a direct correlation between the DAT in the abovementioned studies and the success rate represented in our case with AF recurrence. See Table 2.

### 3.3. Early CA Studies (DAT Time of Less than Three Years)

Early CA studies had a DAT of less than three years. In this category, we found four articles in total, including Andrade et al. [11], Wazni et al. [14], Kuniss et al. [10], Kalman et al. [20], only RCTs, with a total of 553 patients evaluated. Three studies [10,11,14] included cryoablation as an ablation type, while ref. [20] mentioned both radiofrequency and cryoablation as operational types used. They investigated paroxysmal AF [10,11,14], while ref. [20] also had patients with PsAF. The mean DAT was 12 months; AF burden was mentioned in two studies, 0.6 ± 3.3 [11] and 0% [10]. AF recurrence had an average of 31.1% of the recurrence rates mentioned in the studies. The left atrial diameter or left atrial size was assessed in [10,11,14]; it ranged between 37.0 and 39.5 mm, being around normal at 41 mm. At the same time, EF was reported in all four studies, ranging between 57% and 62.8%, respectively. In conclusion, we did not find a direct correlation between the variation of DAT in the studies mentioned above and the success rate represented in our case with AF recurrence. See Table 3.

### 3.4. Mixed DAT Studies

In this category, we found one study by Kawaji et al. [21] with 1206 patients evaluated, which tended to mention both types of DAT in the form of a comparison between two groups of patients. The study included radiofrequency as an ablation type and investigated paroxysmal and PsAF. The DAT varied throughout the subgroups. AF recurrence was not quantifiable in our case since the classification of DAT in the study is different. The left atrial diameter was 40.9 mm, around normal 41 mm. While EF was reported in the study, it was constant at 63.1%. In conclusion, we could not precisely assess the success rate in such a study due to differences in the parameters examined. See Table 4.

### 3.5. Comparison Between Different Approaches in DAT

The DAT varied through each subgroup. In the early ablation group, DAT was between 8.4 and 15.6 months, while in the delayed ablation subgroup, it was between 36 and 51.3 months. After a follow-up period of one year, the early ablation subgroup had a lower mean AF recurrence rate of 29.8% (95% CI: 17–42.7), with substantial heterogeneity (I^2^ = 90.36%, *p* < 0.001) compared to that of the delayed ablation subgroup 39.5% (95% CI: 29.6–49.5). A leave-one-out analysis was performed due to high heterogeneity, but it showed that no singular study had a major impact on the results of the analysis. See Figure 3a,b.

### 3.6. Comparison Between AF Recurrences in Paroxysmal AF Studies

In the paroxysmal AF subgroup, early ablation was associated with a pooled one-year recurrence proportion of 26.8% (95% CI: 12.4–41.2) across three studies (*n* = 464), with substantial heterogeneity (I^2^ = 92.5%, *p* < 0.001). Leave-one-out analysis identified Andrade et al. [11] as the primary source of heterogeneity; its exclusion reduced heterogeneity to 0% and yielded a more consistent pooled recurrence rate of 19.3% (95% CI: 14.9–23.7) across the remaining two studies (*n* = 310). In contrast, delayed ablation studies (*n* = 638) reported a higher pooled recurrence proportion of 31% (95% CI: 25.4–36.6) with moderate heterogeneity (I^2^ = 42.0%, *p* = 0.189). These findings suggest that, within paroxysmal AF populations, early ablation, particularly when excluding outlier effects, may be associated with a lower one-year recurrence rate compared to delayed intervention. See Figure 4a–c.

### 3.7. Comparison Between AF Recurrences in Different Ablation Procedures

In the following Figure 5, we compared the studies analyzed in different groups depending on the ablation procedure used. As a primary result, we observed that the mean AF recurrence rate at one-year follow-up in the radiofrequency ablation group was 44.5% (95% CI: 37.8–53.3), with substantial heterogeneity (I^2^ = 89.62%, *p* < 0.001) in comparison to the AF recurrence rate in the cryoablation group, which was 27.3% (95% CI: 18.2–36.4) with I^2^ = 89.62% and a *p* < 0.001. A leave-one-out analysis was performed, but it showed that no singular study had a major impact on the results of the analysis.

### 3.8. Assessment of Follow-Up Methods Used

In the delayed ablation group, the main follow-up method was ECG, followed by Holter, physical examination with a scheduled meeting, and TTM (target temperature management). In the early ablation group, the main follow-up method used was Holter, followed by ECG and implantable device monitoring, such as an implantable loop recorder. In the mixed subgroup, both Holter and implantable devices were the main follow-up methods.

## 4. Discussion

Our results have shown that DAT correlates with a recurrence rate at one year following AF CA. The shorter the DAT, the better the outcome, particularly in the paroxysmal AF population.

### 4.1. Definition of Early and Delayed Rhythm-Control Strategy

Several studies declare that AF is the most common sustained arrhythmia. AF forces the progress from paroxysmal AF (AF episodes lasting <7 continuous days) to PsAF (AF episodes lasting ≥7 continuous days) by the process of remodeling, and this could be electrical, contractile, and structural remodeling. Studies reported that AF progression at 10 years follow-up is 22 to 36% of patients and at 12 months is 8 to 15% of patients [22,23,24,25]. According to Kalman et al. [20] The concept of the time between AF diagnosis and first AF ablation, or DAT, was created by observational studies, systematic reviews, and registries [16,18,25,26]; it is crucial and seems to be an important factor and has an impact on the ablation success. The optimal timing of CA is still unknown, and there is a heterogeneous definition of early and delayed CA. A study by Nastasa et al. suggests that a good cutoff value for DAT is 2 years [27]. The EAST-AFNET4 [28], an RCT, tested whether a strategy of early rhythm-control therapy that included CA impacts clinical outcomes in early AF patients; they defined an early rhythm-control strategy within the first year after diagnosis. This definition was similar to the median time of diagnosis to ablation in the first-line cryoballoon CA trials [10,11,14]; from this definition of what an early rhythm control means, Kalman et al. [20] defined a delayed ablation strategy as being at the 12-month mark and early ablation within one month. This definition of delayed ablation contradicts the previous RCT that defined an early ablation strategy within the first year after diagnosis [29]. Maurizio et al. [19] also divided patients into an early CA group within 15 months or less from AF diagnosis and a late CA group of more than 15 months. On the other hand, three observational studies reported a DAT of 39 months [16], four years [17], and a mean DAT of 47 months [26]. A summary of how the included studies defined early versus delayed ablation is presented in Table 5. The analyzed studies revealed significant variations in baseline characteristics, including age, sex, comorbidity burden, and AF subtype. While Hussein et al. [18] focused on patients with persistent AF and a high prevalence of comorbidities with a 24-month follow-up, whereas Kuniss et al. [10] studied a younger cohort with paroxysmal AF and minimal comorbidities with a 12-month follow-up. These differences in patient profiles, ablation techniques, and monitoring methods introduce variability that affects recurrence rates and limits comparability. Failing to account for this heterogeneity may obscure conclusions about the impact of DAT. A more focused discussion of these differences would enhance the findings interpretative value and emphasize the need for individualized patient selection and standardized reporting in future trials.

### 4.2. Correlation and Impact Between DAT and Treatment Outcomes

In studies focusing on paroxysmal AF, the recurrence rate of AF was found to be directly proportional to DAT. These results should primarily apply to paroxysmal cases until more specific research on persistent AF becomes available. The findings of this study predominantly involve cohorts with paroxysmal AF, raising questions about the generalizability of these results to patients with persistent AF. Clinicians should therefore exercise caution in interpreting the data concerning its implications for persistent AF, highlighting the need for further research directed specifically at this population. In this context, Hussein AA et al. [18] conducted a study examining the outcomes of PVI ablation in relation to DAT. This study encompassed 1241 patients with PsAF who underwent CA and discovered that the recurrence rates of arrhythmia progressively increased with longer DAT, rising from 33.6% to 52.6%, 57.1%, and 54.6%, respectively. Additionally, longer DAT was significantly associated with a degree of atrial remodeling. These findings indicate a strong and direct correlation between DAT, atrial remodeling, and the recurrence of arrhythmias.

Moreover, the one-year follow-up period may not adequately capture the long-term effectiveness of various ablation strategies. In many of the studies included, the follow-up duration was limited to just one year. It may prove insufficient for thoroughly assessing the long-term efficacy and durability of different ablation techniques. Given that atrial fibrillation is a progressive condition, recurrences can occur beyond the one-year mark, potentially leading to an underestimation of the true differences in outcomes between early and delayed interventions.

AF recurrence rate in the cryoablation group was lower (27.3%) in comparison to the radiofrequency ablation group (44.5%) at 1-year follow-up. This technical imbalance makes it difficult to determine whether the observed benefits of early intervention truly reflect optimal timing or simply the inherent advantages of cryoablation in less remodeled atria. An imbalance in ablation modalities used across studies may have influenced the observed outcomes. Early intervention groups often underwent cryoballoon ablation, which is known to have favorable outcomes in less remodeled atria. This raises the question of whether the benefits seen with early ablation truly reflect the impact of timing or are partly attributable to the inherent procedural advantages of cryoablation. Disentangling these effects will require future studies that control for ablation technique. Recently, a meta-analysis of over 2200 patients demonstrated that pulsed field ablation (PFA) led to lower rates of arrhythmia recurrence, shorter procedural times, and a reduced risk of phrenic nerve injury, despite a slightly higher risk of pericardial tamponade in some cases [30]. The MANIFEST-PF registry reported favorable one-year arrhythmia-free survival rates of 78.1%, with low rates of serious complications (≤1.9%) [31]. Furthermore, PFA eliminates the need for contact-force monitoring and real-time temperature management, streamlining the procedural workflow. However, recent biochemical analyses have raised concerns about increased markers of hemolysis and myocardial injury following PFA compared to RF ablation, although these findings have not yet translated into meaningful differences in clinical outcomes [32]. As such, while PFA represents a promising advancement in AF therapy, further randomized, long-term studies are warranted to validate its safety, durability, and broader applicability across AF subtypes.

Per the 2023 ACC/AHA/ACCP/HRS guideline update, catheter ablation is recommended as first-line rhythm-control therapy (Class I) in selected patients with symptomatic paroxysmal atrial fibrillation, particularly younger individuals with minimal atrial remodeling, to improve symptoms and reduce progression to persistent AF. These recommendations reflect growing evidence that earlier ablation, performed before substantial structural or electrical remodeling occurs, can yield better outcomes. For patients with device-detected or subclinical AF, current guidelines emphasize the importance of individualized management and shared decision-making, as the prognosis can vary based on the duration of episodes, symptom burden, and comorbidities. This raises the possibility that for such patients, if there is progression of AF, an increasing burden of arrhythmia, or the emergence of symptoms, considering earlier ablation may be beneficial. Additionally, a subgroup analysis that categorizes patients based on both their arrhythmia burden/duration and their CHA2DS2-VASc score could provide valuable insights into which groups are most at risk [33]. AF duration is associated with progressive structural and electrical remodeling of the atrial substrate, as demonstrated by contemporary imaging and electrophysiological modalities. Late gadolinium enhancement cardiac MRI (LGE-MRI) and electroanatomic voltage mapping (EAM) reveal that longer AF duration correlates with increasing left atrial fibrosis and the expansion of low-voltage areas, indicative of substrate degradation [34,35,36]. Structural remodeling not only predisposes individuals to the initial onset of AF but also increases the risk of recurrence. It is evident sometimes even in asymptomatic populations, but also serves as a robust predictor of arrhythmia recurrence following catheter ablation [34,37]. Quantitative fibrosis assessment using LGE-MRI, such as the Utah staging system (Stages I–IV), provides both prognostic and mechanistic insight into AF progression [34,38]. In parallel, electrical remodeling, reflected in parameters such as atrial fibrillation cycle length (AFCL) and bipolar voltage, mirrors these structural changes, reinforcing the link between AF burden and the evolving atrial substrate [36,39,40]. Collectively, these data highlight the value of integrating advanced imaging and electroanatomic mapping to stratify risk and guide personalized therapeutic strategies in AF management. Emerging evidence also suggests that AF burden, as detected by continuous monitoring through implantable cardiac devices, may be a more accurate predictor of AF progression and recurrence than AF history duration alone, highlighting the importance of rhythm surveillance in clinical risk stratification [41,42].

## 5. Conclusions

Our results suggest that DAT correlates with a recurrence rate at one year following AF CA. A limit of three years of DAT between early and delayed ablation may improve outcomes following AF ablation, particularly in patients with paroxysmal AF.

## 6. Limitation

Five of our included studies were observational, which cannot provide strong evidence, such as a randomized clinical trial design. Some of the studies lacked a comparative medical treatment arm and missed asymptomatic recurrences. Furthermore, not all studies used implantable loop recorders, which provide continuous cardiac monitoring; instead, ECG and Holter monitoring may underdiagnose the incidence of atrial tachyarrhythmias. It is important to note that there is an inhomogeneous aspect to the results between early CA studies and delayed CA studies. The three-year threshold for defining early versus delayed ablation is a data-driven choice based on the distribution of ablation timing in our study. While this cutoff is consistent with some prior literature, we must acknowledge the potential arbitrariness of this threshold and encourage future studies to explore and validate alternative definitions. Studies with long-term follow-up were associated with high AF recurrence rates, whereas studies with low documented AF recurrence rates had short-term follow-up, which is in favor of early CA. In other words, the length of the follow-up period delivers valuable information about the natural history of AF. Additional studies are needed to address the longer-term safety and clinical effectiveness. Overall, this review has provided quantitative, synthesized evidence and supports the benefit of early intervention, particularly in the paroxysmal population, so generalizability is limited.

## Figures and Tables

**Figure 1 jcm-14-06995-f001:**
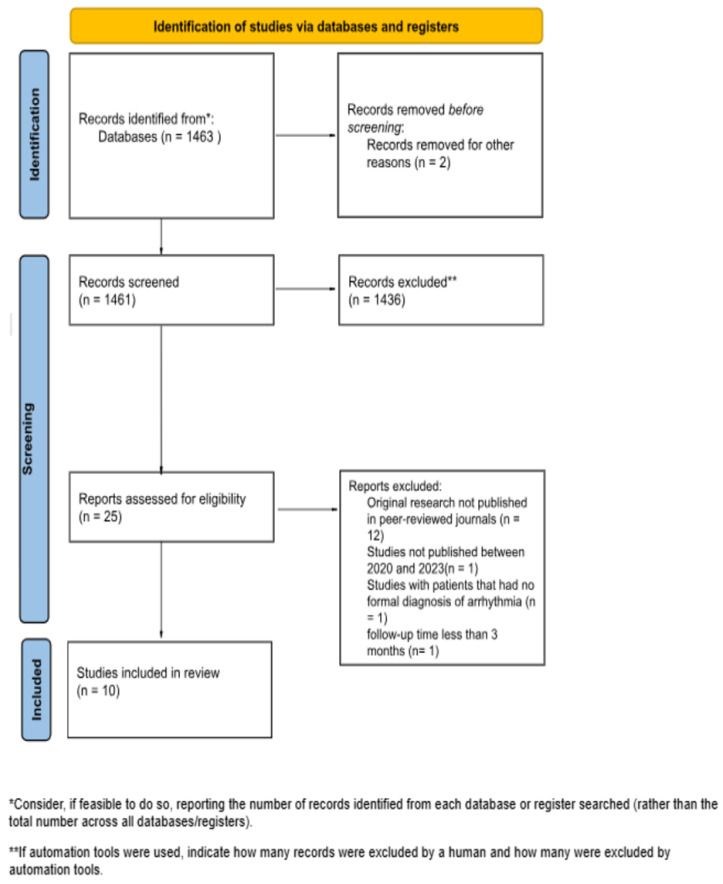
PRISMA study selection flow diagram.

**Figure 2 jcm-14-06995-f002:**
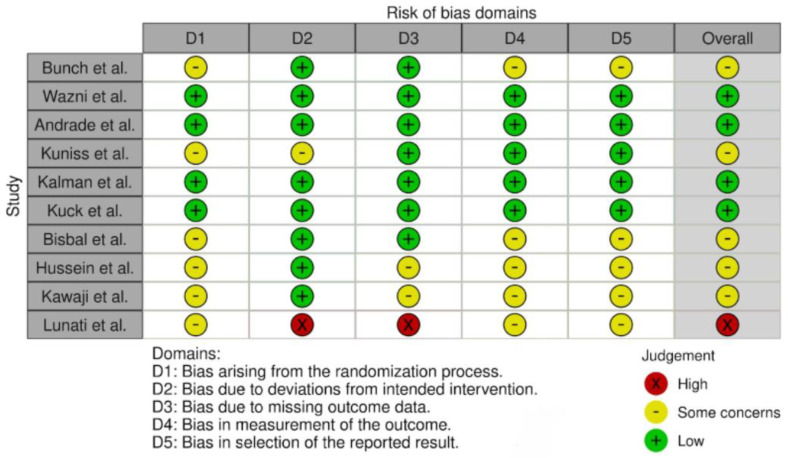
Risk-of-bias domains: ROB-2 ROB-2: Risk-of-bias tool [10,11,12,14,16,17,18,19,20,21].

**Figure 3 jcm-14-06995-f003:**
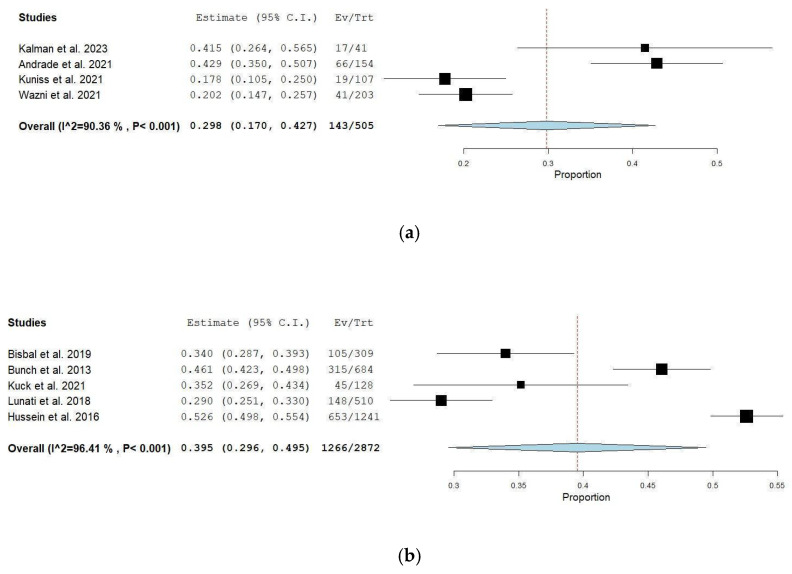
(**a**) Overall early studies [10,11,14,20]. (**b**) Overall delayed studies [12,16,17,18,19].

**Figure 4 jcm-14-06995-f004:**
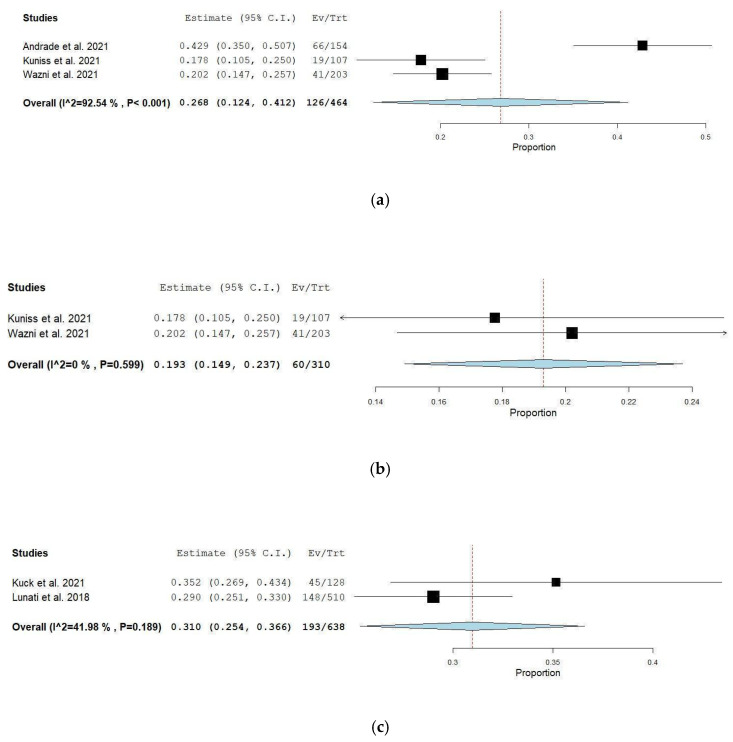
(**a**) Early all paroxysmal AF subgroup [10,11,14]. (**b**) Post-exclusion early paroxysmal AF subgroup [10,14]. (**c**) Delayed paroxysmal subgroup [12,19].

**Figure 5 jcm-14-06995-f005:**
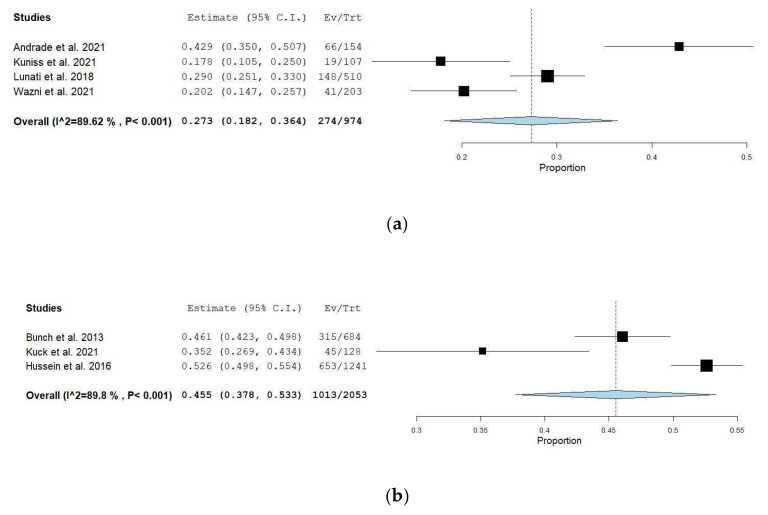
(**a**) Cryoablation studies [10,11,14,19]. (**b**) Radiofrequency ablation studies [12,16,18].

**Table 1 jcm-14-06995-t001:** Patient’s characteristics.

Study	Number of Patients	Mean Age	Male Sex *n* (%)	Hypertension *n* (%)	Diabetes *n* (%)	Dyslipidemia *n* (%)	Heart Failure *n* (%)	CHA_2_DS_2_-VAScmean ± SD	Obesity (%)/BMI ± SD
Bunch et al., 2013 [16]	684	65	419 (61)	512 (74.8)	117 (17.1)	214 (31.3)	203 (29.7)	Most < 3	NR
Kuck et al., 2021 [12]	128	67.7	54 (42.2)	120 (93.8)	13 (10.2)	67 (52.3)	24 (18.8)	NR	NR
Bisbal et al., 2019 [17]	309	56.9	219 (71)	138 (44.5)	26 (8.4)	NR	NR	2.0 ± 1.5	Obesity (21.3) BMI 28.1 ± 4.9
Andrade et al., 2021 [11]	154	57.7	112 (72.7)	57 (37.0)	NR	NR	14 (9.1)	1.9 ± 1.0	Obesity (36.4) BMI 30.9 ± 14.2
Wazni et al., 2021 [14]	203	60.4	63 (61)	58 (56)	15 (14)	NR	1 (1)	Most < 3	NR
Kuniss et al., 2021 [10]	107	50.5	76 (71)	33 (30.8)	1 (0.9)	23 (21.5)	0 (0.0)	Majority 0–1	NR
Hussein et al., 2016 [18]	1241	61	974 (78.5)	478 (38.5)	105 (8.5)	NR	NR	NR	NR
Lunati et al., 2018 [19]	510	59.1	344 (67.5)	241 (47.2)	26 (5.1)	NR	NR	Majority 0–2	BMI 27 ± 4
Kalman et al., 2023 [20]	89	59	Early 32 (67) Delayed 31 (76)	Early 26 (54) Delayed 22 (54)	Early 3 (6) Delayed 1 (2)	NR	Early 6 (13) Delayed 2 (5)	Early (48) 1.8 ± 1.6 Delayed (41) 1.2 ± 1.1	BMI Early (48) 30 ± 5.5 Delayed (41) 30 ± 5.3
Kawaji et al., 2019 [21]	1206	64.3	856 (71)	707 (58.6)	195 (16.1)	NR	102 (8.5)	2.0 ± 1.5	NR

NR: not reported.

**Table 2 jcm-14-06995-t002:** Delayed CA studies (DAT of three years or more).

	Year Published	Type of Study	Ablation Strategy	Parox/Pers	Mean DAT	Mean/Median Follow-Up (Time)	Mean AF Burden	AF Recurrence	Left Atrium Diameter (mm)/Size (cm^2^)	EF
Delayed ablation studies (DAT ≥ 3 years)	Bisbal et al., 2019 [17]	Observational prospective	Cryo + RF	66.8% Parox	51.3 months	13.7 months	NR	34.2%	41.5	61.2%
	Lunati et al., 2018 [19]	Observational prospective	Cryo	Parox	36 months	16.3 months	NR	29%	40.8	60.2%
	Bunch et al., 2013 [16]	Observational retrospective	RF	30–180 days group (56.1% parox.) 181–545 days (58.6% parox) 546–1825 days (57% parox) > 1825 days (59.5% parox)	39 months	39.4 months	NR	30–180 days (38.1%) 181–545 days (45.7%) 546–1825 (47.8%) >1825 (46.0%)	Not reported	53%
	Hussein et al., 2016 [18]	Observational prospective	RF	Pers	36 months	24 months	NR	48.3%	26.7	52.3%
	Kuck et al., 2021 [12]	Randomied clinical trial (RCT)	RF	Parox	51.2 months	36 months	NR	49.2%	42.7 ± 5.9	NR

NR: not reported, AF: Atrial Fibrillation, Cryo: Cryoballoon Ablation, DAT: Diagnosis-to-Ablation Time, EF: Ejection Fraction, Parox: Paroxysmal Atrial Fibrillation, Pers: Persistent Atrial Fibrillation, RF: Radiofrequency Ablation.

**Table 3 jcm-14-06995-t003:** Early CA studies (DAT Time of less than three years).

	Year Published	Type of Study	Ablation Strategy	Parox/Pers	Mean DAT	Mean/Median Follow-Up (Time)	Mean AF Burden	AF Recurrence	Left Atrium Diameter (mm)/Size (cm^2^)	EF
Early Ablation (DAT < 3 years)	Andrade et al., 2021 [11]	Randomized clinical trial (RCT)	Cryo	Parox 95.5%	12 months	12 months	0.6 ± 3.3	42.9%	39.5 ± 5.0	59.6 ± 7.0
	Wazni et al., 2021 [14]	Randomized clinical trial (RCT)	Cryo	Parox	15.6 months	12 months	NR	20%	38.7 ± 5.7	60.9 ± 6.0
	Kuniss et al., 2021 [10]	Randomized clinical trial (RCT)	Cryo	Parox	8.4 months	12 months	NR	17.8%	37.0 mm	62.8
	Kalman et al., 2023 [20]	Randomized clinical trial (RCT)	Cryo + RF	Parox 46%	One month (early), 12 months (delayed)	12 months	(Early vs. Delayed: 0% vs. 0%)	43.7% -> early ablation arm 41.4% delayed arm	NR	in early -> 57 ± 8, in delayed -> 61 ± 5

AF: Atrial Fibrillation, Cryo: Cryoballoon Ablation, DAT: Diagnosis-to-Ablation Time, EF: Ejection Fraction, Parox: Paroxysmal Atrial Fibrillation, Pers: Persistent Atrial Fibrillation, RF: Radiofrequency Ablation, NR: not reported.

**Table 4 jcm-14-06995-t004:** Mixed DAT studies.

	Year Published	Type of Study	Ablation Strategy	Parox/Pers	Mean DAT	Mean/Median Follow-Up (Time)	Mean AF Burden	AF Recurrence	Left Atrium Diameter (mm)/Size (cm^2^)	EF
Mixed DAT Studies	Kawaji et al., 2019 [21]	Observational retrospective	RF	Parox 70.7%	short (less than 36 months) (N = 675) and long (more than 36 months) (N = 531)	60 months	NR	38.6% (short DAT) 29.9% (long DAT)	40.9	63.1%

AF: Atrial Fibrillation, DAT: Diagnosis-to-Ablation Time, EF: Ejection Fraction, Parox: Paroxysmal Atrial Fibrillation, Pers: Persistent Atrial Fibrillation, RF: Radiofrequency Ablation, NR: not reported.

**Table 5 jcm-14-06995-t005:** Definitions of early versus delayed ablation across studies.

Study	DAT
Bunch et al., 2013 [16]	39 months (delayed)
Kuck et al., 2021 [12]	51.2 months (delayed)
Bisbal et al., 2019 [17]	51.3 months (delayed)
Hussein et al., 2016 [18]	36 months (delayed)
Lunati et al., 2018 [19]	36 months (delayed)
Andrade et al., 2021 [11]	12 months (early)
Wazni et al., 2021 [14]	15.6 months (early)
Kuniss et al., 2021 [10]	8.4 months (early)
Kalman et al., 2023 [20]	1 month (early), 12 months (delayed)
Kawaji et al., 2019 [21]	<36 months (early) >36 months (delayed)

DAT: Diagnosis-to-ablation time.

## Data Availability

No new data were created or analyzed in this study. Data sharing is not applicable to this article.

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
