# Peer review of "Correlation Between Catheter Ablation Timing and the Duration of Atrial Fibrillation History on Arrhythmia Recurrence in Patients with Paroxysmal Atrial Fibrillation: A Systematic Review and Meta-Analysis"

_jcm, 2025, doi:10.3390/jcm14196995_

Round 1

Reviewer 1 Report

Comments and Suggestions for Authors

I have read the manuscript of work „Correlation between catheter ablation timing and the duration of atrial fibrillation history on arrhythmia recurrence in patients with paroxysmal atrial fibrillation: A systematic review and meta-analysis..”

Authors demonstrated that ten studies were selected and after a follow-up period of 1 year, the early ablation subgroup had a lower mean AF recurrence rate (29.8%) compared to that of the delayed ablation subgroup (39.5%). The median AF recurrence rate was in the radiofrequency ablation group (44.5%), in the cryoablation group (27.3%). In studies that included paroxysmal AF patients exclusively, the AF recurrence rate was directly proportional to the DAT. In conclusion results study suggest that DAT correlates with recurrence rate at 1 year following AF CA, the shorter the DAT the better the outcome, particularly in paroxysmal AF population.

The subject of work is interesting and work is correct writing.

I propose to supplement the discussion with the American guidelines.

Author Response

We thank the reviewer for their positive evaluation of our work and valuable suggestion. In response, we have supplemented the Discussion section by incorporating relevant recommendations from the latest American guidelines (2023 ACC/AHA/ACCP/HRS) on atrial fibrillation management. This addition provides a comprehensive clinical context for our findings, particularly regarding the timing of catheter ablation and its role in rhythm control strategies.

We believe this enhancement strengthens the manuscript by aligning our results with contemporary guideline-directed practice and improving its clinical applicability. The relevant text can be found between lines 392–406 in the revised manuscript.

Reviewer 2 Report

Comments and Suggestions for Authors

Limitations

  • Five of the included studies were observational.
  • Absence of comparative arms with medical therapy.
  • Monitoring was not always continuous (lack of implantable loop recorders).
  • Arbitrary definition of “early” and “delayed” ablation timing.
  • Follow-up duration was often short.

Strengths of the Study

  • Systematic and rigorous methodological approach.
  • Quantitative analysis using robust statistical methods.
  • Focus on a clinically relevant and underexplored topic (timing of catheter ablation).
  • Good stratification of subgroups (AF type, ablation modality, diagnosis-to-ablation time).

Critical Issues

  • High heterogeneity among the included studies.
  • Predominance of observational designs.
  • Inconsistent definitions of “early” and “delayed” ablation across studies.
  • Follow-up is limited to one year in most cases.
  • Potential confounding between ablation modality and timing (cryoablation is more frequently used in early intervention groups).

Final Judgment on Publishability

The study is well-structured, methodologically sound, and addresses a clinically relevant question. Despite some limitations (heterogeneity, arbitrary definitions, short follow-up), the results are consistent and supported by appropriate statistical analyses. Authors provided certain methodological aspects (e.g., definition of DAT) that need to be clarified and the discussion of limitations further strengthened

Author Response

We sincerely thank the reviewer for the comprehensive and constructive evaluation of our manuscript. We appreciate the recognition of our systematic approach, robust statistical analyses, and the clinical relevance of our study. We have acknowledged and addressed the limitations raised. These changes can be found between lines 379–425. Thank you again for your insightful comments, which have significantly strengthened our work.

Reviewer 3 Report

Comments and Suggestions for Authors

Congratulations to the authors for their manuscript that deals with one of the hottest topic in current electrophysiologic literature. I just have some comments about it:

Although the results indicate a correlation between atrial fibrillation history and arrhythmia recurrence, the manuscript provides limited discussion of the underlying pathophysiological mechanisms. The authors are encouraged to expand the discussion by incorporating evidence from contemporary electrophysiological and imaging studies, such as late gadolinium-enhancement cardiac MRI and electroanatomic voltage mapping, which elucidate the relationship between AF duration and structural or electrical remodeling of the atrial substrate.

The  definitions of terms such as “early ablation” and “prolonged AF history” vary considerably across the included studies. To enhance clarity it would be helpful to include a table or supplementary figure that summarizes how each study defines these key thresholds.

The authors should elaborate on the clinical implications of their findings, particularly in relation to guideline-directed AF management and the timing of referrals for catheter ablation especially in patients with subclinical or device-detected AF. It would also be valuable to reference emerging literature suggesting that AF burden, as detected by implantable cardiac monitors, may be a more meaningful predictor of recurrence and progression than AF history duration alone. The authors cannot ignore the latest evidences in AF ablation literature, such as the groundbreaking impact of pulsed field ablation; authors are then encouraged, in order to make their manuscript up to date, to include the latest evidences of such ablation energy compared to radiofrequency (doi: 10.1111/jce.16776)

Author Response

We sincerely thank the reviewer for this thorough and insightful feedback, which has greatly helped us improve the clarity, depth, and contemporary relevance of our manuscript. In response, we have:

  • Expanded the Discussion to incorporate recent electrophysiological and imaging evidence — including late gadolinium-enhancement cardiac MRI and electroanatomic voltage mapping — that elucidates the pathophysiological relationship between AF duration and atrial substrate remodeling and emerging prognostic importance of AF burden measured by implantable monitors. (Lines 406-425)

  • Substantially elaborated on the clinical implications of our findings in the context of current guideline-directed AF management, with particular attention to optimal referral timing for catheter ablation in patients with subclinical or device-detected AF. (Lines 393-406)

  • Updated the manuscript to include the most recent advances in AF ablation, highlighting the groundbreaking potential of pulsed field ablation. (Lines 379-391)

We believe these revisions significantly enhance both the scientific content and the clinical applicability of our work. All changes are detailed in the revised manuscript between lines 379–425.

Thank you again for your insightful comments, which have significantly strengthened our work.

Round 2

Reviewer 3 Report

Comments and Suggestions for Authors

my congratultions to the authors for the revised version of their manuscript. 

Author Response

We are truly appreciative of the reviewer’s positive assessment of our revised manuscript. The supportive remarks, along with the constructive suggestions provided throughout the review process, have been invaluable in refining and enhancing the quality of our work.